# Climate Effects on Ergot and Ergot Alkaloids Occurrence in Italian Wheat

**DOI:** 10.3390/foods13121907

**Published:** 2024-06-17

**Authors:** Mariantonietta Peloso, Gaetan Minkoumba Sonfack, Ilaria Prizio, Eleonora Baraldini Molgora, Guido Pedretti, Giorgio Fedrizzi, Elisabetta Caprai

**Affiliations:** 1Food Chemical Department, Istituto Zooprofilattico Sperimentale della Lombardia e dell’Emilia Romagna (IZSLER), Via Fiorini 5, 40127 Bologna, Italy; m.peloso@izsler.it (M.P.); g.minkoumbasonfack@izsler.it (G.M.S.); ilaria.prizio@izsler.it (I.P.); e.baraldinimolgora@izsler.it (E.B.M.); giorgio.fedrizzi@izsler.it (G.F.); 2Agronomist Freelance, 22100 Como, Italy; guido.pedretti@gmail.com

**Keywords:** ergot alkaloids, sclerotia, climate change, durum wheat, mycotoxins

## Abstract

In recent years, there has been an intensification of weather variability worldwide as a result of climate change. Some regions have been affected by drought, while others have experienced more intense rainfall. The incidence and severity of moldy grain and mycotoxin contamination during the growing and harvesting seasons have increased as a result of these weather conditions. Additionally, torrential rains and wet conditions may cause delays in grain drying, leading to mold growth in the field. In July 2023, a wheat field in Lecco (Lombardy, Italy) was affected by torrential rains that led to the development of the *Claviceps* fungi. In the field, dark sclerotia were identified on some ears. Wheat ears, kernels, and sclerotia were collected and analyzed by LC-MS/MS at IZSLER, Food Chemical Department, in Bologna. The wheat ears, kernels, and sclerotia were analyzed for 12 ergot alkaloids (EAs) according to (EU) Regulation 2023/915 (ergocornine/ergocorninine; ergocristine/ergocristinine; ergocryptine/ergocryptinine; ergometrine/ergometrinine; ergosine/ergosinine; ergotamine/ergotaminine), after QuEChERS (Z-Sep/C18) purification. The analyzed sclerotia showed significant differences in total alkaloid content that vary between 0.01 and 0.5% (*w*/*w*), according to the results of the 2017 EFSA scientific report. EAs detected in sclerotia were up to 4951 mg/kg, in wheat ears up to 33 mg/kg, and in kernels were 1 mg/kg. Additional mycotoxins, including ochratoxin A, deoxynivalenol, zearalenone, fumonisins, T2-HT2 toxins, and aflatoxins, were investigated in wheat kernels after purification with immunoaffinity columns (IAC). The analysis revealed the presence of deoxynivalenol in wheat kernels at a concentration of 2251 µg/kg. It is expected that climate change will increase the frequency of extreme weather events. In order to mitigate the potential risks associated with mycotoxin-producing fungi and to ensure the protection of human health, it is suggested that official controls be implemented in the field.

## 1. Introduction

Climate change has significantly contributed to numerous weather and climate extreme events in recent years. Observations have shown changes in extreme events such as heatwaves, heavy rainfall, droughts, and tropical cyclones. The frequency and intensity of these events have increased in most regions of the world since the 1950s, reducing food security due to warming [1]. Agriculture is highly vulnerable to adverse climate events; therefore, climate change is expected to increase uncertainties in providing sufficient and safe food for the growing world population of the 21st century, with major impacts on crop productivity and quality [2]. Mycotoxin contamination is a significant risk due to heavy rains and wet conditions, which could delay grain drying and promote mold growth in fields. One of the most widespread mycotoxigenic species affecting cereals in the European field is *Claviceps* spp., which produces ergot alkaloids (EAs) [3]. In particular, *Claviceps purpurea* parasitizes the seed heads of living plants at the time of flowering and commonly affects cereals such as rye, wheat, triticale, barley, millet, and oats [4].

The fungus replaces the grains on cereal ears or seeds, forming visible sclerotia, known as “ergot” from the French word “argot” meaning cock’s spur. The typically purple or black sclerotia fall to the ground before or during harvest and remain intact through winter and grain storage [5,6].

The impact of weather on sclerotial germination and host infection is significant (e.g., summer rainfall). The increasing frequency of weather-related stress may also affect pollen viability, particularly in cereals such as wheat and barley [7].

*Claviceps purpurea* produces more than 50 different alkaloids. European Food Safety Authority (EFSA) report of 2012 identified ergotamine, ergocristine, ergosine, ergocornine, ergocryptine, ergometrine, and their corresponding -inine epimers as the most important alkaloids to monitor [6].

The alkaloid content of a sclerotium varies considerably, ranging from 0.01% to 0.5% (*w*/*w*), and the alkaloid profile is determined by the fungal strain and host plant in a given geographical region [5].

The EAs are classified as ergopeptines and ergopeptinines, which have different biological and physicochemical properties. These alkaloids have a stereocentre in position C8, which exists in an R- (suffix: -ine) or S- (suffix: -inine) configuration (Figure 1) [6,8,9].

The -inine forms are considered to be biologically inactive, although some of them may rapidly epimerize under certain conditions, such as heating, acidic solvents, or changes in pH. However, epimerization is a reversible process in which the two epimers seek thermodynamic equilibrium [4,10]. On the other hand, the -ine form is considered to be more active in terms of toxicity. Therefore, to avoid underestimation of the total biologically active EA, EFSA suggested that both forms should be quantified [6,11].

Ergotism is the oldest known mycotoxicosis, caused by devastating epidemics in humans resulting from the consumption of food contaminated with EAs [12]. In the Middle Ages, mainly in Central Europe, an ergot alkaloid intoxication, or ergotism, was known as St. Anthony’s Fire or Holy Fire [5] from *ignis sacer* because of the burning sensations in the limbs [13]. The disease appeared in two characteristic forms: convulsive ergotism, characterized by muscular tremors, convulsions, hallucinations, and gangrenous ergotism, with severe vasoconstriction leading to limb necrosis and autoamputation [14].

Concerns about these mycotoxins led to the setting of new maximum levels. In 2021, the first European Commission Regulation 2021/1399 was implemented, setting maximum levels for sclerotia of *Claviceps* spp. and for 12 EAs in cereals and derived products for human consumption. This regulation was repealed in 2023 by (EU) Commission Regulation 2023/915 [15,16].

Between 2020 and 2024, there was a significant increase in RASFF (Rapid Alert System for Food and Feed) notifications related to ergot alkaloid contamination in several European countries. Specifically, there were eight notifications for the presence of sclerotia and 12 for the presence of EAs. The notifications, mainly from Germany, concerned food and feed containing rye [17].

However, ergot alkaloids are increasingly found in other cereals such as triticale, barley, wheat, and oats [18], probably due to climate change, as the occurrence of these alkaloids and other mycotoxins in winter cereals is associated with more frequent rainfall and wet soils during critical periods [18,19].

This study examines the case of a wheat field in Lombardy that was lodged by heavy summer rains (Figure 2) and in which *Claviceps* sclerotia were found.

The wheat ears and kernels, including the sclerotia, were collected and analyzed by LC-MS/MS after QuEChERS (quick, easy, cheap, effective, rugged, and safe) (Z-Sep/C18) purification. Furthermore, the sample of wheat kernels was analyzed for other mycotoxins (aflatoxins, ochratoxin A, deoxynivalenol, zearalenone, fumonisins, and T2-HT2 toxins) by LC-MS/MS after separation and purification with immunoaffinity columns (IAC).

### Case Report

This study examines a case that affected a mountain wheat field in La Valletta Brianza, located in the province of Lecco (Lombardy, Italy) (Figure 3), during the summer of 2023. Lombardy experienced heavy summer rains, causing the lodging of durum wheat ears in July 2023. The soil was affected by the development of a microclimate with high humidity and low ventilation, which made the wheat more susceptible to infection by *Claviceps* fungi.

*Claviceps purpurea* generally infects the host plant during two critical periods: the rainy and the flowering seasons [12]. The fungus germinates during cold weather followed by warm periods (0–18 °C). Within 24 h, *Claviceps* spores germinate and infect the host [20,21,22].

The data on temperatures, precipitation, and humidity relative to the growing and harvesting period of wheat (March–July 2023) were collected from the meteorological station closest to the field, which was contaminated by the *Claviceps* spp. (Montevecchia Cascina Butto meteorological station, Lecco) [23]. The data are compared with historical data of temperature and precipitation in the area (available data 1991–2023 Vertemate con Minoprio (CO) meteorological station, 30 km from the contaminated field) and historical data of relative humidity (available for the Bergamo, Lombardy, meteorological station, 1993–2023), as shown in Appendix A.

Reports from the Regional Agency for Services to Agriculture and Forestry (ERSAF) [24] indicate that the durum wheat flowered in May, that it was a particularly rainy month in Lombardy, with humidity levels of 85%, above historical data (Appendix A). This may have contributed to the development of fungal infections. In June, the grain filling stage continued high humidity levels, further increasing fungal infection risks and affecting grain quality and safety. It requires 4–6 weeks after infection for sclerotia to form and replace the seed [20,21,22]. In July, the infection expanded as rainfall increased, and visible dark sclerotia were found on wheat ears at harvest time in weed-infested ground (Figure 4). In the field, a type of weed, *Convolvulus* spp., was observed, which tied and held the wheat ears to the ground following the lodging caused by the rains and strong winds.

## 2. Materials and Methods

### 2.1. Sampling

The sample was collected by an agronomist in a wheat field located in La Valletta Brianza (Lecco), in the northern Italian region of Lombardy. It was collected from 50% of the cultivated field area, corresponding to 1400 m^2^.

It consisted of wheat ears embedding ergot sclerotia, single sclerotia dropped to the ground (Figure 5), and semi-mature wheat kernels. The sample was separated at the laboratory and analyzed as the following: (1) whole wheat ears (*n* = 11), (2) Sclerotia selected from each ear (*n* = 11), (3) whole single sclerotia collected from the ground (*n* = 4), and (4) wheat kernels sample (2 kg).

### 2.2. Chemicals and Reagents

All reagents and solvents were of analytical grade. Certified reference standards of ergosine, ergosinine, α-ergocryptine, α-ergocryptinine, ergocornine, ergocorninine, ergocristine, ergocristinine, ergometrine, ergometrinine, ergotamine, and ergotaminine were purchased from Biopure (Romer Labs, Tulln, Austria).

Methanol was purchased from VWR Chemicals (Rosny-sous-Bois, France), formic acid and acetonitrile were from Carlo Erba Reagents (Val de Reuil, France), and ultrapure water was from Evoqua Water Technologies (Pittsburgh, PA, USA). Ammonium carbonate was purchased from Merck (Darmstadt, Germany), and ammonia 25% and Z-Sep/C18 QuEChERS tubes were from Supelco (St. Louis, MO, USA).

Standard stock solutions were prepared by dissolving reference materials in acetonitrile to give a concentration of 100 µg/mL for each -ine form and 25 µg/mL for each -inine epimer. A working standard solution containing all 12 EAs at 0.1 µg/mL was prepared in acetonitrile. A calibration curve ranging from 0.03 to 5 ng/mL was prepared in methanol/water/formic acid 60:40:0.4 (*v*/*v*/*v*). The sample extraction solution was 5 mM ammonium carbonate/acetonitrile 15:85 (*v*/*v*).

### 2.3. Sample Preparation

The durum wheat sample was analyzed for the presence of *Claviceps* alkaloids. Wheat kernels were finely milled, and a slurry mixing was prepared (sample/water ratio 1:1 *v*/*v*). An aliquot, 10 ± 0.1 g, of the homogenized sample was weighted and extracted with 40 mL of extraction solution, vortexed, horizontal shaken for 30 min, and centrifuged at 4000 rpm at 4 °C. An aliquot portion of the extract was withdrawn and cleaned up with Z-Sep/C18 QuEChERS. After centrifugation, the supernatant was nitrogen-dried at ambient temperature, reconstituted with 0.5 mL methanol/water/formic acid (60:40:0.4 *v*/*v*/*v*), and further diluted 1:50 with the same solvent. The analysis was conducted on two replicates.

Single wheat ears and ergot sclerotia were individually ground and entirely extracted with 10 mL of extraction solution. After shaking and centrifugation, the supernatant was cleaned up with Z-Sep/C18 QuEChERS, nitrogen dried and reconstituted with 0.2 mL methanol/water/formic acid (60:40:0.4 *v*/*v*/*v*), and further diluted 1:100 with the same solvent to have a response within the calibration range.

### 2.4. LC-MS/MS Analysis

The ergot alkaloids analysis was performed using a liquid chromatography system consisting of an Acquity I class UPLC (Waters) coupled to a Xevo TQxS mass spectrometer. Mobile phase A was 10 mM ammonium carbonate solution (pH 10 with ammonia 25%), while mobile phase B was acetonitrile. EAs separation was achieved by means of an Acquity UPLC BEH C18, 1.7 µm, 2.1 mm × 100 mm column using a gradient-elution chromatography, as detailed in Appendix A.

In order to highlight characteristic transitions and the most appropriate detection conditions, tuning solutions of individual analytes were directly infused into the mass spectrometer detector. For each precursor ion, two daughter ions were monitored [25]. The LC-MS/MS parameters for all analytes are presented in Table 1. The fragmentation was carried out in the positive ionization mode (ESI +) under the conditions reported in Appendix A.

### 2.5. Quantification

According to SANTE/12089/2016 [25], identification and quantification of ergot alkaloids was carried out on the basis of analytes retention time, ion fragments and ion ratio and compared to those originating from reference standard and control samples (blank samples fortified with EAs at limit of detection value). Specification for ionic ratio was ±30% and ±0.1 min for the retention time.

The matrix effect was assessed prior to validation procedures and evaluated as not relevant. All EAs had an instrumental response in a matrix-matched diluent that did not differ by more than 20% from that of a solvent.

For this reason, ergot alkaloids concentration was extrapolated by means of the least squares regression method based on a calibration curve prepared in methanol/water/formic acid 60:40:0.4 (*v*/*v*/*v*) ranging from 0.03 to 5 ng/mL. LC-MS/MS chromatograms of 12 EAs at a concentration of 3 ng/mL are shown in Appendix A.

### 2.6. Performance Evaluation

Further to the European Commission Recommendation 2012/154/EU, the analytical method for monitoring the occurrence of ergot alkaloids in food has been developed and validated [26]. In accordance with the EURL-MP guidance on performance criteria for plant toxins and Regulation 2782/2023/EC [27,28], the following performance parameters were evaluated: specificity, recovery rates, linearity, repeatability, intra-laboratory reproducibility, and limit of quantification (LOQ). The method can be applied to cereals and similar products, and it is linear for concentrations of EA between 2 and 600 µg/kg. The linearity of the reference materials was confirmed by a correlation coefficient (R^2^) higher than 0.99 and residuals ≤20%. Specificity was verified by analyzing blank matrices such as rye flakes, oat flakes, oatmeal, wheat, wheat flour, and barley. No significant interference peaks were identified in the chromatograms, and these food matrices were used for validation purposes. Repeatability and recovery were assessed by analyzing the blank samples spiked with EAs at different concentrations in six replicates per level. Spiking values were 2, 10, 50, 150, 250 and 600 µg/kg. According to the EURL-MP guidance [28], for each concentration level, the mean recovery for each ergot alkaloid and for the sum of the 12 of them was in the range of 70–120%. Repeatability, expressed as percentage relative standard deviation (RSDr), was ≤20% (Appendix A). Intra-laboratory reproducibility (RSD_R_) was ≤20% and was calculated on-going by assessing the outcomes of six batch sequences under routine conditions as recommended by EURL-MP guidance [28]. The LOQ was set at 2 µg/kg as stated in Regulation 2782/2023/EC [27].

### 2.7. Other Mycotoxins Investigation

The wheat kernels slurry was analyzed in two replicates for additional mycotoxins, including aflatoxins, ochratoxin A, deoxynivalenol, zearalenone, fumonisins, and T2-HT2 toxins. Analytes were extracted with organic solvents and purified with IAC. The analyses were carried out using validated and accredited LC-MS/MS analytical methods routinely applied by the laboratory for official controls.

The LC-MS/MS parameters for all mycotoxins analyzed are presented in Appendix A. The fragmentation was carried out in the positive ionization mode for all mycotoxins (ESI +), except for zearalenone (ESI−), under the conditions reported in Appendix A. LC conditions and gradient-elution program are showed in Appendix A.

## 3. Results and Discussions

### 3.1. Ergot Alkaloids Occurrence

The wheat sample analyzed for EAs in this study showed that the alkaloid content in sclerotia was highly variable, ranging from 0.01 to 0.5%, with no correlation between alkaloid content and individual sclerotia size (Figure 6), as confirmed by the 2017 EFSA report [29].

The total content of EAs (T-EAs), which refers to the sum of 12 EAs, ranged from 3 to 4951 mg/kg in sclerotia, up to 33 mg/kg in wheat ears, and 1 mg/kg in wheat kernels sample (Appendix A). As shown in Figure 7, the wheat ears were visibly affected by *Claviceps* fungus. Regulation 915/2023 [15] sets maximum residue limits (MRLs) based on the sum of the epimeric forms. This implies that epimerization is not critical as long as the EAs sum remains unchanged. However, the EAs were also individually quantified. The study revealed that the alkaloid profile was very similar: all 12 ergot alkaloids were consistently present in the samples. Additionally, the levels of ergotamine/ergotaminine were the lowest, as shown in Appendix A. Ergosine was the predominant alkaloid in 33% of the sclerotia samples, with concentrations ranging from 92.8 to 705.7 µg/kg. Ergometrine was the most frequently detected alkaloid in 72% of the wheat ear samples, with concentrations ranging from 8 to 55 µg/kg. In the kernels, ergocristine was the most prevalently detected alkaloid, with a value of 350 µg/kg.

According to other European studies [29,30,31,32], in analyzed samples, the -ine (R)-epimer forms were much higher than the -inine forms (92% and 8%). The epimerization occurs spontaneously, but its mechanisms remain largely unknown [9]. In the case of cereal-derived products, such as bread, epimerization may depend on the processing of the raw material, such as baking [29,30].

### 3.2. Mycotoxins Occurrence

Temperature, humidity, water availability, and insects are among the factors that can affect the infection of plants by mycotoxic fungi and the production of mycotoxins. In addition to *Claviceps* spp., several other mycotoxigenic species may infect the crops, including *Fusarium*, *Aspergillus,* and *Penicillium*. It is worth noting that *Fusarium* species are known to commonly colonize crops in the field, while *Aspergillus* and *Penicillium* species are more likely to grow on cereals in unsuitable conditions during drying and storage [2].

In this study, the wheat kernels sample was also analyzed for other mycotoxins (aflatoxins, ochratoxin A, deoxynivalenol, zearalenone, fumonisins, and T2-HT2 toxins) (Appendix A) and only deoxynivalenol (DON) was found at a concentration of 2251 µg/kg, which exceeds the maximum level of 1750 µg/kg for unprocessed durum wheat grains, for human consumption, set by EU Regulation 2023/915 [15].

In another recent Italian study [33], high levels of DON have been found in wheat kernels, and its occurrence has always been associated with extreme climatic conditions, such as heavy rainfall during the flowering season. Furthermore, recent quantitative estimates suggest that cereals in certain regions of Europe may experience increased contamination with DON and aflatoxin B1 due to global warming [2,34,35].

### 3.3. Potential Contamination Scenarios

The results of this study showed that the concentration of ergot alkaloids detected in sclerotia varied independently of their size, as detailed in Appendix A. These results were used to formulate possible contamination scenarios in order to assess the potential contamination of food products, such as 1 kg of durum wheat flour, produced from wheat kernels milled in the presence of sclerotia.

Three scenarios were evaluated, as summarized in Table 2, based on different sclerotia weights and on the lowest and highest EA concentrations observed for each type of sclerotia. The data available in this study permitted the estimation of the total amount of EAs that might contaminate 1 kg of durum wheat flour.

The EU Regulation 2023/915 states that the total EAs content in wheat products for human consumption should be between 100 (50 µg/kg from 1 July 2024) and 150 µg/kg, depending on the ash content [15]. In the worst-case scenario (scenario 1, Table 2), the accidental occurrence of milled sclerotia together with wheat kernels might lead to flour contamination with EAs content of 231 µg/kg, exceeding the legal limits. It is worth noting that the situation is more concerning for infant foods, where the limits are significantly lower (20 µg/kg).

These scenarios are based on the analysis results of wheat samples collected from the field after heavy rainfall without undergoing any cleaning procedures [29].

However, it is important to highlight that cleaning procedures do not always effectively reduce the EA contamination in food. For example, small sclerotia may grow to kernel size during drought conditions, or intact sclerotia may break into small fragments during transport [5,36].

### 3.4. Climate Change and Ergot Alkaloids Occurrence in European Countries

The Copernicus Climate Change Service (C3S), implemented by the European Centre for Medium-Range Weather Forecasts (ECMWF), has published its latest annual reports on the European State of the Climate (ESOTC). The reports highlight that Europe has experienced warmer-than-average climate scenarios in recent years, with the mean temperature in 2020 being more than 1.6 °C above the average. The year 2021 was colder than previous years but still warmer than the average. In 2022, the temperature was 0.9 degrees above the average, which may have contributed to a decrease in rainfall. In 2023, Europe experienced its second-warmest year on record, with temperatures 1.02 °C above the average. The summer saw above-average precipitation over most of Western Europe, resulting in local rainfall records being broken and leading to flooding in some cases [37].

Mycotoxins are typically not found in food or feed from European countries. However, changes in the distribution of certain target fungal species in regions affected by climate change may lead to their occurrence. For instance, a greater increase in the distribution of *Fusarium* fungi and mycotoxins is expected in EU countries [38].

In the summer of 2023, Italy experienced widespread and intense storms and rainfalls, particularly in northern Italy [39], which resulted in the proliferation of *Claviceps* spp. and *Fusarium* spp. in the field, as described in this study.

The intoxication caused by the consumption of products derived from cereals contaminated with ergot alkaloids reached its peak in Europe in the Middle Ages when many thousands of people were affected by ergotism [13].

The latest recorded epidemics were in Germany (1879–1881), Ethiopia (1977–1978), Russia (1926–1927), France (1951), India (1975), Brazil (1995), and Australia (1996), leading to increased controls [5,13,40,41]. Climate change has brought higher temperatures, with intense summer rainfall alternating with periods of drought, leading to the growth of the fungus, making it an ongoing problem to be solved. Ergot contamination is still a major problem for agriculture [5,13].

Over the last century, precipitation has increased by 10–40% in Northern Europe and decreased in Southern Europe.

The average annual temperature in Europe is projected to increase by 2.0 °C to 6.3 °C over the next century. Studies have shown that climate change will significantly affect mycotoxin contamination in various cereals [2,42,43,44].

EFSA has examined the impact of climate change on the occurrence of mycotoxins in certain regions. While it may have a negative effect in some areas, it could also be beneficial for specific geographical regions [45,46].

From 2020 to 2024, twenty-two RASFF notifications have been generated mostly from North European countries due to an increase in EAs contamination in food and feed [17]. Germany has a higher frequency of ergot contamination, as evidenced by the number of alerts confirmed by several studies [5,11,47].

EFSA’s 2017 report on dietary exposure to ergot alkaloids (EAs) in humans and animals identified an increase in their occurrence across 22 European countries between 2004 and 2016. The report also revealed that *Claviceps* spp. contamination was highest in the rye and its derivatives. The presence of sclerotia was linearly correlated with EAs content; however, measurable levels of EAs were still found even in the absence of sclerotia [29].

In recent years, several European countries have reported cases of ergot contamination in cereals (Table 3). In Belgium, a 2012 study [48] showed that EA levels in contaminated cereal samples ranged from 1 to 1145 µg/kg. However, a more recent 2021 study analyzed 49 samples of cereal-based baby food from the Belgian market and found that all samples complied with the MRL ≤ 20 µg/kg [49].

In a 2018 study, ergot alkaloids were found in all 122 samples of rye grains harvested in Poland, with levels ranging from 4.7 to 667.9 μg/kg [31].

An Albanian study found that EA levels in winter wheat samples in 2014 ranged from 17.3 to 975.4 µg/kg and from 10.3 to 390.5 µg/kg in 2015 [50], indicating significant contamination.

In a Dutch study, 2020, bread made from different cereals (wheat, rye, multigrain, rye and wheat) showed EAs ranging from LOQ (0.3–1.2 µg/kg) to 335 µg/kg [51].

In Slovenia, 206 wheat samples were analyzed in 2020. The EAs content ranged from 14 to 4217 µg/kg, with a mean value of 363 µg/kg. The highest EA concentrations were found in 16 oat samples (84 to 2191 µg/kg) with a mean value of 594 µg/kg [52].

In other Italian studies, the situation is comparable to our study. Debegnach et al. (2019) observed that 87% of analyzed wheat or rye flour and bread samples from the Italian market were contaminated. EA levels ranged from 2.5 to 1142.6 µg/kg; the most contaminated food category was wheat bread [30]. Lattanzio et al. (2021) reported a range of 2.7–270.7 µg/kg of EAs in 67 cereal samples collected in central and southern Italy between 2017 and 2020 [53]. Carbonell-Rozas et al. (2023) demonstrated the correlation between heavy precipitation and *Claviceps* spp. contamination in cereals. The study analyzed different types of cereals grown side by side and harvested between 2020 and 2022 in Northern Italy. In 2020, the overall contamination of all considered species was particularly high, with 60% of positive samples being found. This was caused by a greater amount of rainfall during the growing season (April–June) compared to other years.

Tritordeum, a hybrid crop obtained by crossing durum wheat with the wild barley *Hordeum chilense* [54], and rye crops were the most contaminated, with levels ranging from 4302 to 15,389 μg/kg [19].

## 4. Conclusions

This work shown how adverse climatic events, dependent on climate change, may negatively affect durum wheat production, leading to the growth of mycotoxigenic fungi that pose a health risk to consumers of cereal-based food.

Climate change can affect fungal infections both directly and indirectly. Directly, it can increase the susceptibility of the host to infection through heavy rains, heat, and drought stress. Indirectly, it can affect the seeding time of crops. Mycotoxigenic fungi may infect crops at any stage of the growing chain, including harvest, transport, and storage. Pre-harvest strategies, including good agricultural practices (GAP), such as rotating out susceptible crops, are often the simplest method of preventive control of ergot bodies.

The study results showed that the content of ergot alkaloids in the analyzed sclerotia was independent of the ergot size. The highest EAs content found in sclerotia weighing 0.03 g was 4951 mg/kg, while in sclerotia weighing 0.2 g, the highest content was 1159 mg/kg. On the basis of the scenarios presented in Table 2, it is estimated that even the smallest sclerotia might contaminate 1 kg of durum wheat flour at a concentration higher than the legal limits set in EU Regulation 2023/915. This is of particular concern for cereal-based baby foods, where the legal limit is much lower at 20 µg/kg.

Consumption of food from cereals contaminated with ergot alkaloids may result in human and animal intoxication, indicating that ergotism remains a relevant risk and has not yet been completely eliminated. However, when food products are already contaminated, decontamination requires physical, chemical, and biological processes, many of which can only reduce the toxicity of ergot alkaloids by promoting the epimerization process.

Cereals are widely recognized as an important food category due to their high nutritional value and their crucial role in a balanced diet. In order to mitigate the occurrence of mycotoxin-producing fungi, it is recommended that all stages of production be closely monitored. It is suggested to implement official controls in the field to protect human health and economic losses.

## Figures and Tables

**Figure 1 foods-13-01907-f001:**
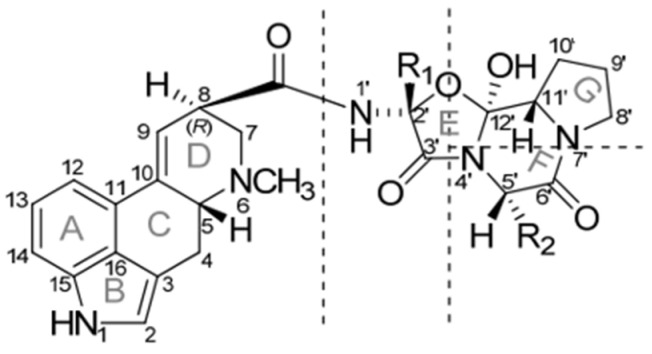
Ergopeptines (or ergopeptides) are composed of lysergic acid and a tripeptide moiety that is condensed into a tricyclic system. R_1_ and R_2_ correspond to the side chain of the amino acids involved; adapted from reference [6].

**Figure 2 foods-13-01907-f002:**
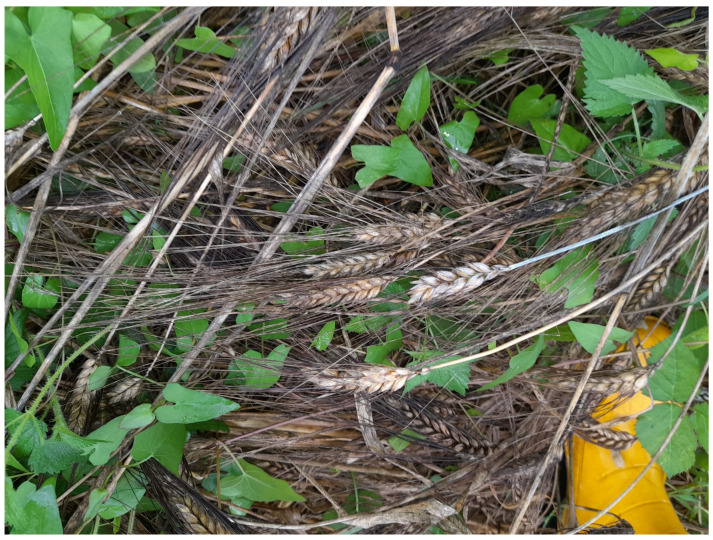
The durum wheat field *Claviceps p.* contaminated (Lombardy, Italy): dark wheat ears lodged.

**Figure 3 foods-13-01907-f003:**
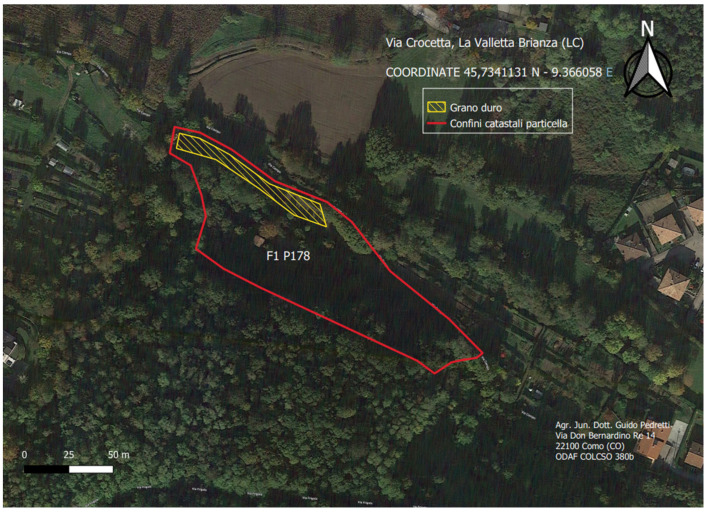
The mountain wheat field in Lombardy, Italy.

**Figure 4 foods-13-01907-f004:**
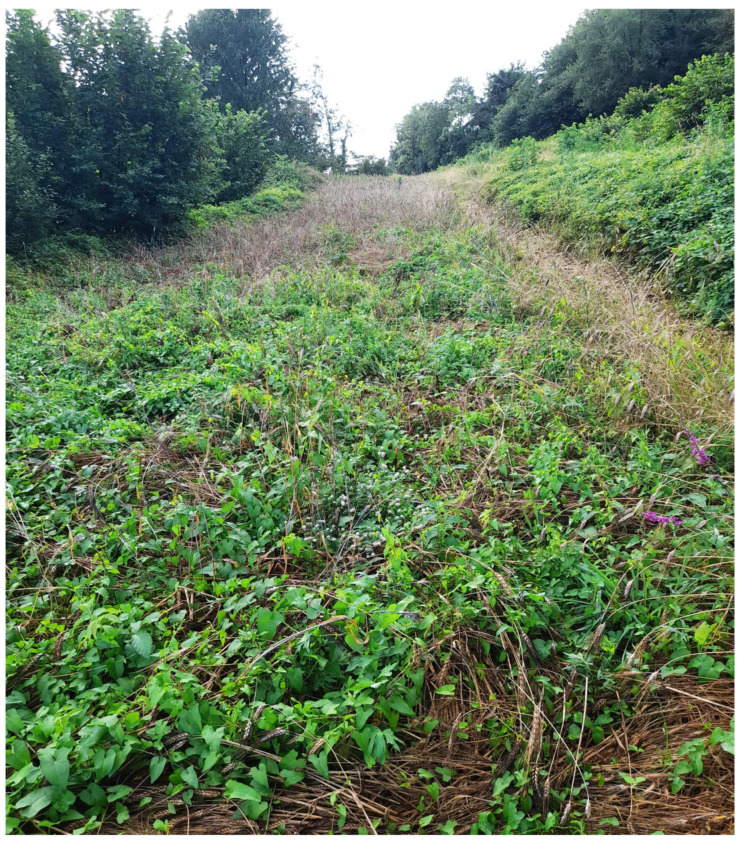
The wheat field in La Valletta Brianza infested with *Convolvulus* spp. (Lecco, Lombardy, Italy).

**Figure 5 foods-13-01907-f005:**
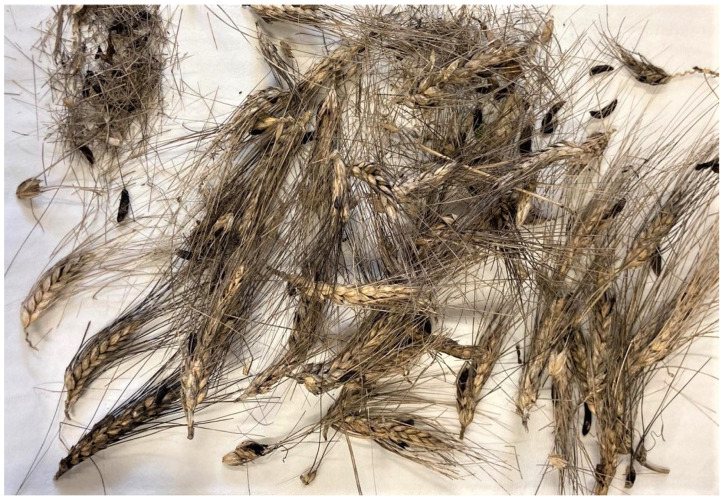
Wheat sample collected in Lecco province (Lombardy, Italy).

**Figure 6 foods-13-01907-f006:**
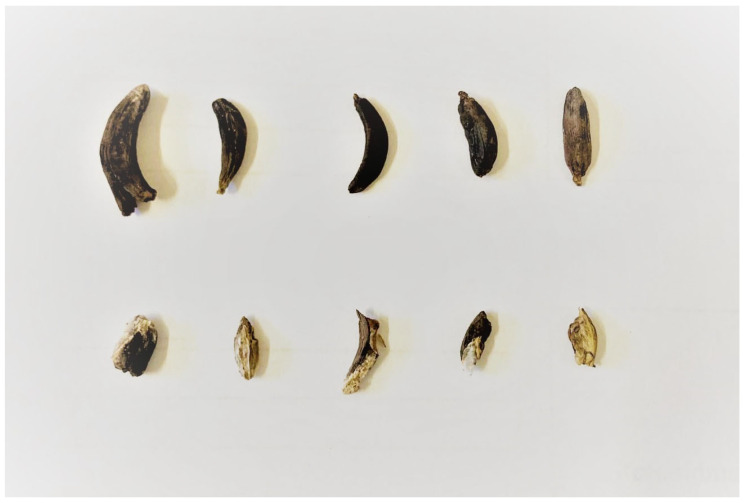
Sclerotia samples.

**Figure 7 foods-13-01907-f007:**
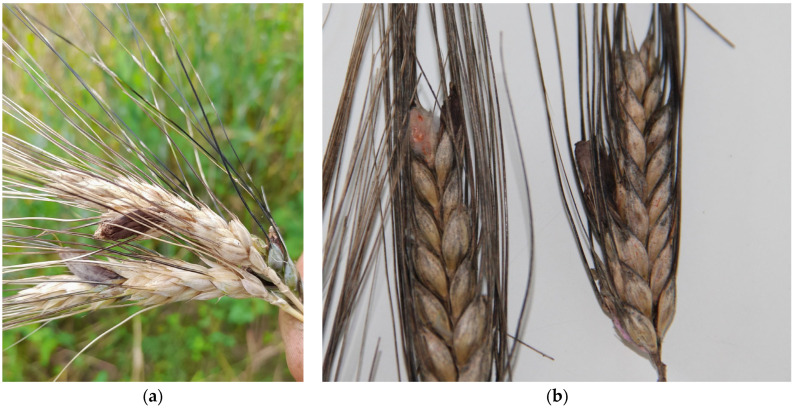
The durum wheat ears contaminated with ergot (**a**,**b**).

**Table 1 foods-13-01907-t001:** LC-MS/MS parameters for all EAs.

Ergot Alkaloids	Retention Time	Dwell Time (s)	Precursor Ion (*m*/*z*)	Product Ion ^1^ (*m*/*z*)	Collision Energy (V)
Ergocornine/ -inine	5.32/6.82	0.020	562.4	223.0/304.9	30/24
Ergocryptine/ -inine	5.72/7.27	0.020	576.4	223.0/305.0	30/24
Ergocristine/ -inine	5.86/7.44	0.020	610.4	223.0/305.0	30/24
Ergometrine/ -inine	1.68/2.77	0.020	326.2	208.1/223.1	30/25
Ergosine/ -inine	4.53/6.11	0.020	548.4	223.0/268.0	28/20
Ergotamine/ -inine	4.77/6.44	0.020	582.4	208.1/223.1	35/40

^1^ Q/q (Q = qualifier, q = quantifier).

**Table 2 foods-13-01907-t002:** Potential contamination scenarios.

Scenario	1	2	3
Sclerotia mean weight	0.20 g	0.10 g	0.03 g
EAs content in sclerotia ^1^	110–1159 mg/kg	147–1422 mg/kg	3–4951 mg/kg
EAs content in food product ^2^	22–231 µg/kg	14–142 µg/kg	0.10–149 µg/kg

^1^ The lowest and the highest EAs content values found for each sclerotia. ^2^ The lowest and the highest EAs content estimated in food products (1 kg of durum wheat flour contaminated after milling with sclerotia).

**Table 3 foods-13-01907-t003:** European research on the occurrence of EAs in cereals.

Country	Year	N. of Analyzed Samples	Sample Types	EAs Range µg/kg	Research
Belgium	2012	122	cereals	1.0–1145	[48]
Belgium	2021	49	cereal-based baby food	≤20.0	[49]
Poland	2018	122	rye grains	4.7–667.9	[31]
Albania	2014–2015	71	wheat	10.3–975.4	[50]
Netherlands	2014–2018	40	cereals bread	0.3–335.0	[51]
Slovenia	2020	206	wheat	14.0–4217	[52]
Italy	2019	71	wheat/rye flour and bread	2.5–1142	[30]
Italy	2021	67	cereals	2.7–270.7	[53]
Italy	2023	162	cereals	4302–15,389	[19]

## Data Availability

The original contributions presented in the study are included in the article/Appendix A, further inquiries can be directed to the corresponding author.

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
