# Peer review of "Climate Effects on Ergot and Ergot Alkaloids Occurrence in Italian Wheat"

_foods, 2024, doi:10.3390/foods13121907_

Round 1

Reviewer 1 Report

Comments and Suggestions for Authors

This manuscript analyses the effects of Ergot and Ergot Alkaloids in Italian wheat and has important implications for risk assessment and control of Ergot and Ergot Alkaloids. It is suitable for FOODS journals, but has some problems

1. There are only Ergot and Ergot Alkaloids in the title, but there is other fungal toxin information in the text. The title should cover the subject matter of the article.

2. Parameters of detection methods for other toxins should be shown.

3、The size of the pictures in the text is not consistent

4. The section on the effects of climate change on toxins is mainly a review of other literature and does not contain the author's own analytical data. If the authors do not have their own research data, this section should be deleted and the title should be revised.

5. In this study, the number of samples and the sampling quality are very small, how to ensure that the samples taken are representative and homogeneous.

Comments on the Quality of English Language

Moderate editing of English language required

Author Response

The authors would like to express their gratitude to the reviewer for their time and attention. The authors' responses to the reviewer's comments are attached. 

Reviewer 2 Report

Comments and Suggestions for Authors

The article presents a case study concerning grain contamination by mycotoxin-producing fungi.

The research is conducted according to standards, although certain reservations may arise regarding the methodology of sample acquisition.

Samples of grain and sclerotia were collected directly from the field, whereas the presence of mycotoxins in grain intended for processing (post-harvest) and in flour (post-processing) should be of interest. Without data from real post-harvest and post-processing stages, the implications for consumer safety remain somewhat unclear.

Nonetheless, the findings contribute to the existing body of knowledge on food safety and highlight the importance of ongoing research in this area.

Below, please find some remarks concerning the article:

Lines # 111 – 116: Readers would like to see details of meteorological data, i.e. temperature, rainfall, etc., with normal values for reference.

Lines # 134 – 140: Please, give details of the area of sclerotia collection, i.e. what was the % of the whole Triticum durum field examined for sclerotia.

Lines # 353 – 365: Probably it would be better to place given examples of EA contamination in a table.

Author Response

(The authors gave the same response as above.)

Reviewer 3 Report

Comments and Suggestions for Authors

This manuscript describes a mycotoxin, specifically ergot alkaloid, contamination in an Italian wheat field affected by torrential rains attributed to climate change. As such, it appears to be a “case study” to highlight the increase in mycotoxin contamination as a result of climate change.

The manuscript is very well written with few, if any, errors or typos. The introduction is thorough in covering previously published work, quoting some 52 references most of which are recent (within the last five years).  The materials and methods are well described and the LC-MS/MS methodology is detailed and appears to be well validated (section 2.7 and supplemental). The results and implication of those results are well discussed (pages 7-10), with all of the raw results consigned to the supplemental. The authors do a good comparison of their results to previous results in their own country and in other countries (page 10 , lines 348-377). Unlike most manuscripts, it is punctuated with good photos (Figures 2-7) illustrating the damage to the wheat, which makes the manuscript more pleasant to read.

The only criticism has to do with the wheat kernel analysis. Were all the wheat kernels pooled and analyzed as one sample? If so, why and why did the authors not have several representative samples? It looks like there is only one wheat kernel sample in the supplemental as well as section 2.5 and page 5, lines 161-162. However, the way it is expressed in the abstract and on page 5, line 140 suggests multiple wheat kernel samples.  Also, was only one section of the contaminated field sampled (see page 5, lines 134-140) as a single sample and if so, why? The author need to clarify their sampling and processing protocols.  

Specific comments:

Abstract, line 17: This suggests several samples of wheat ears, kernels and sclerotia.

Page 5, line 140: This statement implies several wheat kernel samples.

Page 5, line 161-162: This description suggests that ALL the wheat kernels were combined and one 10 g aliquot was taken and extracted.

Page 6, section 2.5: If this was only one sample that was analyzed for other mycotoxins, it should read “The wheat kernel sample was”. If there was more than one “the wheat kernel samples were”.

Page 8, lines 268-272: This statement implies more than one wheat kernel sample was analyzed. If that is the case, the raw data should be reported in the supplemental. If this is just one wheat kernel sample, there was just one DON measurement made (and a rather high one at that!) and line 268 should read “the wheat kernel sample was also analyzed for other mycotoxins”…….

Page 8, line 287: This sentence is confusing – what was also included? Please clarify.

Author Response

(The authors gave the same response as above.)

Round 2

Reviewer 1 Report

Comments and Suggestions for Authors

This manuscript has been revised in the light of the comments, and he should have been accepted.

Comments on the Quality of English Language

Minor editing of English language required